# Severe COVID-19 ARDS Treated by Bronchoalveolar Lavage with Diluted Exogenous Pulmonary Surfactant as Salvage Therapy: In Pursuit of the Holy Grail?

**DOI:** 10.3390/jcm11133577

**Published:** 2022-06-21

**Authors:** Barbara Ruaro, Paola Confalonieri, Riccardo Pozzan, Stefano Tavano, Lucrezia Mondini, Elisa Baratella, Alessandra Pagnin, Selene Lerda, Pietro Geri, Marco Biolo, Marco Confalonieri, Francesco Salton

**Affiliations:** 1Department of Pulmonology, Cattinara Hospital, University of Trieste, 34127 Trieste, Italy; paola.confalonieri.24@gmail.com (P.C.); riccardo.pozzan@asugi.sanita.fvg.it (R.P.); stefano.tavano95@gmail.com (S.T.); lmondinifr@gmail.com (L.M.); pietrogeri@gmail.com (P.G.); marcobiolo@gmail.com (M.B.); marco.confalonieri@asugi.sanita.fvg.it (M.C.); francesco.salton@gmail.com (F.S.); 2Department of Radiology, Cattinara Hospital, University of Trieste, 34127 Trieste, Italy; elisa.baratella@gmail.com; 3Pineta del Carso, Viale Stazione, 26, 34011 Aurisina, Italy; alex_pag@yahoo.it; 424ore Business School, Via Monte Rosa, 91, 20149 Milano, Italy; selenelerda@gmail.com

**Keywords:** COVID-19, respiratory failure, non-invasive respiratory support (NIRS), acute respiratory distress syndrome (ARDS), coronavirus disease, surfactant

## Abstract

Background: Severe pneumonia caused by coronavirus disease 2019 (COVID-19) is characterized by inflammatory lung injury, progressive parenchymal stiffening and consolidation, alveolar and airway collapse, altered vascular permeability, diffuse alveolar damage, and surfactant deficiency. COVID-19 causes both pneumonia and acute respiratory distress syndrome (COVID-19 ARDS). COVID-19 ARDS is characterized by severe refractory hypoxemia and high mortality. Despite extensive research, the treatment of COVID-19 ARDS is far from satisfactory. Some treatments are recommended for exhibiting some clinically positive impacts on COVID-19 patients although there are already several drugs in clinical trials, some of which are already demonstrating promising results in addressing COVID-19. Few studies have demonstrated beneficial effects in non-COVID-19 ARDS treatment of exogenous surfactant, and there is no evidence-based, proven method for the procedure of surfactant administration. Aim: The aim of this work is to underline the key role of ATII cells and reduced surfactant levels in COVID-19 ARDS and to emphasize the rational basis for exogenous surfactant therapy in COVID-19 ARDS, providing insights for future research. Methods: In this article, we describe and support via the literature the decision to administer large volumes of surfactant to two patients via bronchoalveolar lavage to maximize its distribution in the respiratory tract. Results: In this study, we report on two cases of COVID-19 ARDS in patients who have been successfully treated with diluted surfactants by bronchoalveolar lavage, followed by a low-dose bolus of surfactant. Conclusion: Combining the administration of diluted, exogenous pulmonary surfactant via bronchoalveolar lavage along with the standard therapy for SARS-CoV-2-induced ARDS may be a promising way of improving the management of ARDS.

## 1. Introduction

Coronavirus disease 2019 (COVID-19), caused by severe acute respiratory syndrome coronavirus 2 (SARS-CoV-2) was declared a pandemic by the World Health Organization (WHO) on 11 March 2020 [1,2,3]. As of 27 May 2022, there have been 524,878,064 confirmed cases of COVID-19, including 6,283,119 deaths, reported to the WHO [1]. COVID-19 causes both pneumonia and acute respiratory distress syndrome (ARDS) [3,4,5,6,7]. The SARS-CoV-2 virus uses the angiotensin-converting enzyme 2 (ACE2) receptor, in conjunction with serine protease TMPRSS2, expressed by alveolar type II (ATII) cells, as one of the SARS-CoV-2 target cells, to enter into the ATII pneumocytes [8,9,10,11,12]. After ATII infection and the resulting damage, the consequences may be severe and may include injury to the alveolar–capillary barrier, lung edema, inflammation, ineffective gas exchange, impaired lung mechanics, and reduced oxygenation, which resembles COVID-19 acute respiratory distress syndrome (COVID-19 ARDS) [10,11,12,13,14,15]. Furthermore, ATII cells are the main source of surfactant. For normal alveolar structure and function, the presence of the right amount of surfactant is imperative for reducing intra-alveolar surface tension. Several publications have demonstrated a reduction in surfactant production in ARDS [16,17,18,19,20,21,22,23]. Few studies have demonstrated beneficial therapeutic effects of exogenous surfactant on gas exchange and lung mechanics in ARDS, and there is no proven method of administration that can maximize its beneficial effect [15,16,17,18,19,20,21,22]. The aim of this paper is to underline the key role of ATII cells and reduced surfactant in COVID-19 ARDS and to emphasize the rational basis of exogenous surfactant therapy in COVID-19 ARDS in our case series.

## 2. Role of ATII Pneumocytes

Alveolar type II (ATII) cells are a key structure of the distal lung epithelium, where they exert their innate immune response and serve as progenitors of alveolar type I (ATI) cells, contributing to alveolar epithelial repair and regeneration. In the healthy lung, ATII cells have several functions, i.e., (a) the transepithelial movement of water and ions that regulates the volume of the alveolar surface liquid, preventing the flooding of the alveoli; (b) the expression of immunomodulatory proteins for host defense; (c) the regulation of innate immunity and the coordination of host defense mechanisms; (d) the regeneration of alveolar epithelium after injury; and (f) the production and secretion of surfactant proteins, which are important for protecting the lungs. Moreover, surfactant proteins help to maintain homeostasis in the distal lung and reduce surface tension at the pulmonary air–liquid interface, thereby preventing atelectasis and reducing the work of breathing [8,9,10,11]. Various acute and chronic diseases are associated with intensive inflammation, such as edema, acute respiratory distress syndrome, and fibrosis. ATII cells may also contribute to the fibroproliferative reaction by secreting growth factors and proinflammatory molecules after damage [8,9,10,11].

## 3. Production and Secretion of Surfactant by ATII Pneumocytes

Type II (ATII) alveolar cells synthesize, store, and release surfactant lipids and proteins in the lungs. ATII cells are unique in their role of synthesizing and assembling all surfactant components (90% lipids and 10% proteins), storing them in specific organelles (lamellar bodies), and finally secreting them by exocytosis into the lumen of the alveolus [8,9,10,11]. Each ATII cell has about 150 lamellar bodies, with an average diameter of 1 μm, and multiple phospholipid bilayers, with a typical unique cellular morphology [8,9,10,11]. Several signaling pathways that regulate surfactant secretion have been extensively characterized. These include 3 distinct signaling mechanisms: (1) the activation of adenylate cyclase, which forms cAMP and activates cAMP-dependent protein kinase; (2) the activation of protein kinase C; and (3) a Ca^2+^-regulated mechanism that probably leads to the activation of Ca^2+^-calmodulin-dependent protein kinase [8,9,10,11,24]. These signaling mechanisms are enabled by a variety of agonists, such as ATP, which activates all three signaling mechanisms. Although the knowledge about the identity of various signaling proteins involved in surfactant secretion is steadily increasing, numerous aspects are still poorly understood. Even if the involvement of the 3 kinases (A, C, and Ca^2+^/calmodulin) in the regulation of surfactant secretion has been established, their physiological substrates have not yet been identified [13,14,15,16]. Furthermore, the role and regulation that the cytoskeleton plays in the fusion of the lamellar body to the plasma membrane and in the recycling of surfactant material, and the identification of the molecular mechanisms that couple the rate of recycling to the rate of secretion, are still debated. These important issues must be better understood if we are to clarify how surfactant secretion and recycling are regulated [11,12,13]. Surfactant lipids have both hydrophilic and hydrophobic properties (amphipathy), and the head groups have charged qualities that form stable, surface-active films at the air–liquid alveolar interface. The low surface tension is provided by phospholipids, predominantly dipalmitoylphosphatidylcholine, and it is assisted by the hydrophobic surfactant proteins SP-B and SP-C [11,12,13]. As only ATII cells produce the surfactant protein C, it is considered an ATII cell-specific marker. Moreover, pulmonary surfactant protects the distal lung from noxious particles and microorganisms [11,12,13]. This surfactant resides on top of a thin water layer that covers the inner surface of the alveoli. Pulmonary surfactant is a complex mixture of phospholipids (PL) and proteins (SP) that reduces surface tension at the air–liquid interface within the alveoli. The pulmonary surfactant that lines the alveoli is essential for life as it lowers alveolar surface tension, preventing atelectasis at the end of expiration. The air–surfactant surface tension varies during the inhalation and exhalation phases. During inspiration, the alveolus expands, resulting in a low surface concentration of surfactant and thus high surface-tension values. At expiration, however, the alveolus contracts, and the surfactant concentration increases, thus reducing the surface tension to near-zero values [11,12,13].

Surfactant production in babies begins around week 24 of gestation and increases rapidly in weeks 34 and 35 (8 months of pregnancy). Hence, premature babies suffer from surfactant deficiency, a condition indicative of the immaturity of the lung, caused by a lack of surfactant, thus developing a respiratory distress syndrome called hyaline membrane disease (HMD). The situation can be traced and successfully treated by the replacement of this substance [16,17,18]. Surfactant deficiency is also observed in adults who suffer from respiratory diseases such as ARDS, and few studies have demonstrated the lack of it in COVID-19 ARDS [12,13,14] as well. One can hypothesize about whether vascular changes lead to substrate depletion in type II cells. It has been experimentally shown that pulmonary microembolization leads to a reduction in the synthesis of lecithin, as well as a reduction in palmitate incorporation, indicating an interruption of the surfactant system [10,16,17,18,19]. It is also conceivable that SARS-CoV-2 can alter surfactant production at the cellular level. Numerous experimental studies have long since proven that, in lung damage such as that resulting from inhaling toxic gases, there are changes in surfactant composition and function. The result of a surfactant dysfunction would be an alveolar collapse [10,16,17,18,19]. A deficiency of pulmonary surfactant results in an interaction between air and water molecules inside the alveoli during inhalation, which, given the different molecular nature of these two components, determines high surface tension. Indeed, during inhalation, water molecules are pushed towards the alveolar surface to create a thinner layer of water. As a response, the alveoli start collapsing after 48–72 h, and damaged cells, known as “hyaline membranes”, accumulate in the airways, causing labored breathing, as well as rattling and bubbling sounds [10,16,17,18,19]. Thus, the existence of the pulmonary surfactant in the alveoli is crucial to reducing the surface tension of the air–liquid interface by limiting the thinning of the layer of water molecules during inhalation and by guaranteeing its natural thickness, which reduces the air–water surface tension.

## 4. ATII in SARS-CoV-2 Infection (COVID-19)

During the SARS-CoV-2 pandemic, the role played by ATII cells was evidenced in the most severe COVID-19 cases. Patients with severe COVID-19 pneumonia suffer from hypoxic respiratory failure, and CT scans show the presence of scattered, subpleural, ground-glass densities during the active viral phase. ATII cells express both angiotensin-converting enzyme 2 (ACE2) and the serine protease TMPRSS2 [10,16,17,18,19].

It is known that freshly isolated ATII cells may vary in their expression of the ACE2 protein and their susceptibility to severe disease [10,16,17,18,19]. The infected ATII cells trigger the innate immune response, which favors virus propagation to adjacent alveoli, as some ATII cells have apical surfaces in more than a single alveolus. The cell-to-cell transmission from ATII to ATI pneumocytes, with viral replication, can rapidly diffuse the damage to the endothelium. The dysregulated inflammatory response causes a cytokine storm, the hallmark of the most severe cases of SARS-CoV-2-related ARDS; this causes fibrinogen and other plasma proteins to leak into the alveoli, where they impair the potentiality that the surfactant has to adsorb to the surface and lower the surface tension. The migration of fibroblast and inflame matory cells into the lumen at the alveolar level causes appositional atelectasis and a loss of gas-exchange units. The ATII cell loss means a loss of progenitors for ATI cells, thus impairing ATI regenerative capacity as well, eventually leading to a progressive worsening of respiratory function [10,16,17,18,19]. After injury, an exaggerated inflammatory response inside the lung, due to the subsequent infection of new areas, may alter the alveolar epithelial cells’ regulation of fibroblast proliferation and the expression of extracellular matrix genes. Moreover, there may also be an alteration of the normal pathways that usually limit fibrosis through the epithelium [10,16,17,18,19].

## 5. COVID-19 ARDS and Treatment by Surfactant Administration

COVID-19 acute respiratory distress syndrome (COVID-19 ARDS) may be considered as a typical example of “primary” ARDS not associated with sepsis, at least in the initial stages. Differently from typical ARDS, in COVID-19, the lung compliance system does not deteriorate rapidly, but gas-exchange impairment and hypoxia predominantly occur due to micro-thrombosis, dead space, and low V/Q lung regions. Associated thrombi can result as expressions of an altered coagulation cascade, which is common in COVID-19 [3]. Severe COVID-19 pneumonia shares several histological and imaging features with non-COVID-19 ARDS since both are characterized by inflammatory lung injury, progressive parenchymal stiffening and consolidation, alveolar and airway collapse, altered vascular permeability, and a pathological picture of diffuse alveolar damage [4]. In the monitoring of severe COVID-19, imaging techniques play a pivotal role. Chest X-rays and computed tomography (CT) scans are useful for diagnosis and clinical management, usually showing the coexistence of typical aspects of viral pneumonia (i.e., ground-glass opacities, with or without consolidations, and crazy-paving opacities, with subpleural distribution) and ARDS features [12].

The latest guidelines for the hospital care of patients affected by coronavirus disease-2019 (COVID-19)-related acute respiratory failure have moved towards a widely accepted use of non-invasive respiratory support (NIRS) in the initial stages of disease. The establishment of severe COVID-19 pneumonia goes through different pathophysiological phases that partially resemble typical ARDS and have been categorized into different clinical–radiological phenotypes. These can variably benefit the application of external, positive, end-expiratory pressure (PEEP) during non-invasive mechanical ventilation, mainly due to varying levels of lung recruitability and lung compliance during different phases of the disease. A growing body of evidence suggests that intense respiratory effort, producing excessive negative pleural pressure swings, plays a critical role in the onset and progression of lung and diaphragm damage in patients treated with non-invasive respiratory support. Routine respiratory monitoring is mandatory to avoid the continuation of NIRS in patients who are at higher risk for respiratory deterioration and who could benefit from non-invasive mechanical ventilation and/or the early initiation of invasive mechanical ventilation. Despite extensive research, the treatment of COVID-19 ARDS is far from satisfactory. Some treatments are recommended for exhibiting some clinically positive impacts on COVID-19 patients although there are also several drugs in clinical trials, some of which are already demonstrating significant promise in addressing COVID-19.

The use of exogenous surfactant in the treatment of ARDS has been investigated in several trials. Some of these studies support its use; however, others have shown that there is no significant improvement. Nine articles were found to support the clinical efficacy of the exogenous surfactant in inflammatory lung diseases: (i) two are pre-clinical trials on animals (rabbits and lambs) [17,19]; (ii) three describe clinical trials on infants and children, including two meta-analyses [25,26], and one comparative study [27]; (iii) three are meta-analyses [20,28] describing the results obtained from several randomized clinical trials on adults; and (iv) one article is a retrospective case-control pilot study [29]. Two articles were found to not support the exogenous surfactant’s clinical efficacy in inflammatory lung diseases in improving mortality and oxygenation for adult ARDS patients [30,31]. Pre-clinical trials on animals showed how exogenous surfactant use resulted in improved lung function and decreased pulmonary edema. In addition to the animal and infant studies, several studies support the use of exogenous surfactant in adults. Three meta-analyses describe the results obtained from several randomized clinical studies [16,20,28]. These studies included the administration of exogenous pulmonary surfactant in adults with acute lung injury and ARDS. These studies have demonstrated beneficial effects in adults affected by ARDS via the administration of exogenous surfactant on gas exchange and lung mechanics, but there is no evidence-based proof for surfactant administration [10,16,17,18,19].

In these particular cases of COVID-19 ARDS, we decided to administer large volumes of surfactant via bronchoalveolar lavage to maximize its distribution in the respiratory tract.

## 6. Clinical Cases

We report the cases of 2 patients with severe acute respiratory distress syndrome (ARDS) due to COVID-19 who have been successfully treated with endobronchial, diluted surfactant administered through a bronchoscope. Both patients required respiratory support with extracorporeal membrane oxygenation (ECMO) and invasive mechanical ventilation due to severe COVID-19-related ARDS at the time of administration. Both patients were in their first day of ECMO support, and the onset of respiratory failure occurred 7 and 8 days earlier, respectively. According to data contained in previously published literature, we chose a commercial, natural surfactant that is routinely used in premature infants (Curosurf, Chiesi farmaceutici SpA, Parma, Italy) [20,28]. Moreover, our decision was supported by the previous positive experience with a young patient, affected by ARDS and Wegener granulomatosis, who benefited from the aforementioned surfactant treatment. A total of 10 vials of Curosurf (each one containing 3 mL of 80 mg/mL surfactant solution, for a total amount of 2400 mg) were diluted with 500 mL of Ringer solution (for a final phospholipid concentration 4.8 mg/mL). The first 250 mL were instilled into the right main bronchus in 5 aliquots of 50 mL during bronchoscopy, and suction was applied after administration. This procedure was repeated 5 times and took about 10 min, while the patient was supine. The left lung was then treated similarly; 207 mL were recovered from both lungs (41.4% of the administered volume). After the lavage procedure, a further bolus of 600 mg of Curosurf was delivered into each main bronchus, and no suctioning was performed after this instillation. The total administered dose of surfactant was 40 mg/kg BW (Figure 1).

Both patients were invasively ventilated under volume-controlled mode, with a tidal volume of 6 mL/kg, according to the current standard for protective ventilation in ARDS, during the whole procedure, and they remained hemodynamically stable. Both patients had been undergoing 80 mg/day methylprednisolone (non-patented drug, ATC code H02AB04) treatment, according to a previously published protocol since hospital admission [32]. They did not undergo any other antiviral or experimental treatment for COVID-19 during their entire hospital stays. The two patients underwent cycles of prone positioning until they were weaned from invasive mechanical ventilation. Both patients were male (60 years old and 66 years old, respectively), active smokers, non-obese (BMI of 23.8 and 26.3, respectively), affected by arterial hypertension and dyslipidemia, but with no other major comorbidities. Neither patient had any other overlapping clinical conditions, besides COVID-19-related ARDS, that could explain the acute respiratory failure.

We focused on the change in lung compliance, oxygenation by pO_2_/FiO_2_ index, and radiological evolution after surfactant therapy (Table 1). A total of 24 h after surfactant treatment, the PaO_2_/FiO_2_ ratio increased from 54.8 to 62.4 in 1 case, and from 61.7 to 72.9 in the other, despite a volume overload of approximately 320 mL. A total of 48 h after treatment, the PaO_2_/FiO_2_ ratios increased to 106 and 98, respectively. A total of 4 days after surfactant treatment, the PaO_2_/FiO_2_ ratios had increased to 136 and 152, respectively. Lung compliance, measured through an inspiratory pause during invasive mechanical ventilation and curarization, rose from 22 to 35 mL/cmH_2_O in one case, and from 28 to 41 mL/cmH_2_O in the other case after 24 h had elapsed since surfactant instillation. In both cases, ECMO support was stopped 6 days after surfactant instillation.

Patients were extubated 13 and 11 days after the surfactant treatment, respectively, and they were discharged from the hospital in good condition after 41 and 38 days, respectively. A total of 3 months after discharge from the hospital, a routine checkup in our outpatient department revealed only a mild reduction in the diffusing capacity for carbon monoxide (DLCO), i.e., 67% and 61% of the LLN (lower limit of normal). A follow-up CT scan 3 months after hospital discharge showed the complete resolution of consolidations, with residual bilateral, low-density ground-glass opacity (GGO) [33,34]. After a further 9 months, the lung function tests and DLCOs were normal, and CT scans demonstrated the complete resolution of GGO (Figure 2).

## 7. Discussion

In our cases of COVID-19-related ARDS, we successfully treated our patients with diluted surfactant by bronchoscopic lavage, followed by a low-dose bolus of surfactant. The patients were extubated after 5 and 8 days of mechanical ventilation, and almost-normal lung function was restored 8 weeks after discharge from our department. Currently, there are neither recommended models for the clinical application nor treatment schedules for the use of surfactant in cases of respiratory failure or ARDS. In clinical studies, surfactant is normally instilled as a bolus, sometimes bronchoscopically on a segmental or lobar level [15]. The bolus method appears to be more efficacious than slow tracheal instillation or aerosol delivery [16,17], but there are disadvantages. First, the distribution of the delivered surfactant in the lung is non-uniform although there is evidence that higher volumes favor the uniformity of distribution [18]. The administration of surfactant via an endotracheal tube does not permit control of the distribution of the surfactant, which may not reach the regions of the lungs that are most severely affected. Second, the patient must be rotated to achieve multiple positions during the application procedure, and this may alter the hemodynamic situation. Third, the method is very expensive because of the large amounts of surfactant needed to overcome the inhibitory effects of serum proteins or blood within the air spaces [7,15]. If we had treated our patients with the bolus method alone (using a minimum of 200 mg/kg), instead of combining lavage with diluted surfactant followed by bolus treatment, the cost of the therapy would have been 5 times higher.

Several pre-clinical and clinical trials show that exogenous pulmonary surfactant has clinical efficacy in inflammatory lung diseases, especially ARDS [17,19,25,26,27,28]. Pre-clinical trials on rabbits and lambs show that pulmonary surfactants improve lung function and reduce the production of pro-inflammatory cytokine inflammation and pulmonary edema, obtaining a significant improvement in oxygenation [17,19]. Clinical trials on infants show that using an exogenous pulmonary surfactant in intubated infants, immediately after birth or after developing ARDS, significantly reduces the occurrence of pneumothorax, pulmonary interstitial emphysema, and neonatal mortality, and it improves the levels of oxygen parameters [25,26]. Randomized clinical trials in adults show that the administration of lung surfactant improves oxygenation in the first 24 h after treatment and appears to reduce the duration of ventilation. The use of this preparation in adults is not associated with reduced mortality or reduced ventilation duration. In our patients, we observed a reduction of ventilation duration. Different studies have reported that surfactant administration has not been shown to improve mortality and oxygenation for adult ARDS. However, several studies report the usefulness of this therapy in children [22,23,29,30,31,32,33,34,35]. We agree with Nakamura et al., who described a case of the bronchoscopic instillation of surfactant in a 9-year-old girl with ARDS and found that this technique allowed surfactant to be given exactly to the desired regions of the lungs [23]. Further, we would suggest combining the application modes and using diluted surfactant to make BAL, with the idea of combining the advantageous effects of large volumes, protein removal, and the lower amounts of surfactant necessary to improve gas exchange [23]. Nakamura et al. used a high dose of surfactant (140 mg/kg BW). Loscar and colleagues described a 19-year-old patient with granulomatosis with polyangiitis (GPA) and ARDS in whom ECMO was initiated because of rapid hemodynamic and respiratory deterioration; after 10 days of ECMO, they ventilated the patient for another 50 days [35]. These workers were able to improve gas exchange and lower the ventilator settings. It is reasonable to assume that our patients’ improvement within the first 24 h after surfactant treatment was due to this intervention. The further clinical course was probably influenced by therapy with prednisolone given 24 h after surfactant treatment. There are no convincing studies demonstrating surfactant dysfunction in COVID-19 patients [36]. However, indirect evidence indicates surfactant disorders in COVID-19 [36,37,38,39]. In fact, we know that SARS-CoV-2 infects the type II alveolar cells that produce surfactant [36,37,38,39]. Furthermore, current evidence supports the benefits of surfactant use in early ARDS [40]. Recently, several groups have undertaken studies to evaluate the potential benefits of exogenous surfactant in COVID-19 patients [40,41,42,43,44].

Furthermore, there are different types of surfactants (e.g., natural bovine and porcine surfactants and synthetic surfactant). A multicenter study showed that early administration of Surfactant-BL (bovine lung extract surfactant) led to reduced mortality in cardiac patients who developed ARDS postoperatively [40,41,42,43,44]. On the contrary, a previous randomized, multicenter trial failed to demonstrate any improvement in mortality and oxygenation following the bolus administration of exogenous natural porcine surfactant to patients with ALI/ARDS. One trial showed that synthetic surfactant containing rSP-C had no clinical benefit to patients with severe, direct lung injury [40,41,42,43,44]. Several meta-analyses showed that all preparations of surfactant similarly failed to reduce mortality. There were insufficient data available for the analysis of changes in oxygenation and ventilation characteristics. We decided to use Curosurf, which is a natural porcine surfactant because it is the only one marketed in Italy. Furthermore, as reported above, we already used Curosurf for another patient, obtaining a good result, and therefore we decided to use it in these two COVID-19 patients.

### Limitations of the Study

There were some limitations to be noted in our study. First, surfactant delivery in our study was performed in a single center, following a uniform protocol with meticulous delivery and retention in the lungs. Second, the major limitation of our study is the small sample size and the absence of controls, which were difficult to enroll due to the extreme severity of the disease in the two patients who underwent surfactant treatment. It was unfortunate that there was insufficient power to statistically confirm the therapeutic role of BAL with diluted surfactant followed by a low-dose bolus in patients with ARDS. Finally, the selection of patients was influenced by the availability of resources, including Curosurf therapy during the pandemic.

## 8. Conclusions

In conclusion, therapeutic BAL with diluted surfactant, followed by a low-dose bolus, seems to be an effective, feasible, and safe method of treating patients with ARDS from different causes. It should be emphasized that the lavage technique using diluted surfactant provides a method of selective and direct drug administration, together with the substantial removal of airway and alveolar debris. Furthermore, the method appears to be very cost-effective. We think that it is the removal of foreign protein, cellular breakdown products, blood, bacteria, and mucus that enhances the therapeutic efficacy of the surfactant and thereby enhances ventilation and gas exchange. We therefore suggest that our case reports could be used to generate hypotheses for future prospective controlled studies, which may prove that diluted surfactant lavage is effective and improves ARDS outcome.

For this reason, a therapeutic strategy in patients with SARS-CoV-2 infection is crucial. An individualized approach based on lung physiology, morphology, imaging, and the identification of biological phenotypes may improve COVID-19 outcomes.

Indeed, the novelty of our study is that we add surfactant therapy in patients with critical COVID-19-related ARDS, opting for a higher dosage than the one reported in the trials, and using the bronchoscope to deposit the surfactant as far as possible in order to get it to the areas where the damage is most relevant and where there is evidence of alveolar collapse. Further research is needed to further analyze the use of the bronchoscope to deposit the surfactant, with the purpose of optimizing the management of patients with ARDS due to COVID-19.

## Figures and Tables

**Figure 1 jcm-11-03577-f001:**
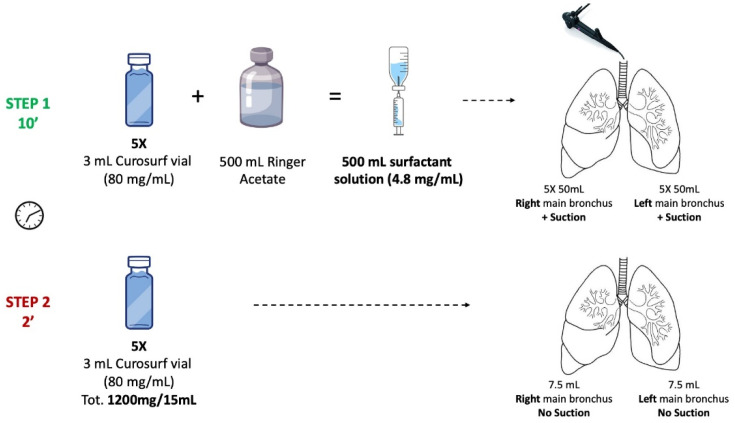
Graphic presentation of the surfactant administration protocol. A total of 10 vials of Curosurf (each one containing 3 mL of 80 mg/mL surfactant solution, for a total amount of 2400 mg) were diluted with 500 mL of Ringer solution (for a final phospholipid concentration 4.8 mg/mL). The first 250 mL were instilled into the right main bronchus in 5 aliquots of 50 mL during bronchoscopy, and suction was applied after administration. This procedure was repeated 5 times and took about 10 min, while the patient was supine. The left lung was then treated similarly; 207 mL were recovered from both lungs (41.4% of the administered volume). After the lavage procedure, a further bolus of 600 mg of Curosurf was delivered into each main bronchus, and no suctioning was performed after this instillation.

**Figure 2 jcm-11-03577-f002:**
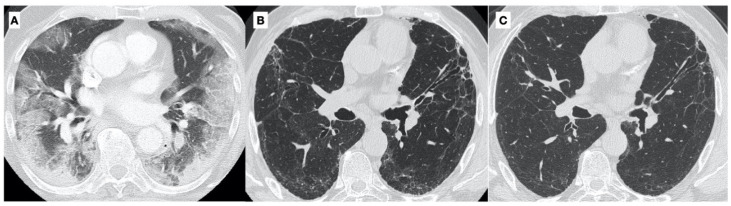
(**A**) Axial High Resolution CT image at baseline in a 60-year-old man (patient n.1) with severe COVID-19 pneumonia, showing bilateral diffuse areas of ground-glass opacities (GGO) and consolidations with peripheral and peribronchovascular distribution. (**B**) A follow-up CT scan 3 months after hospital discharge shows the complete resolution of consolidations, with residual, bilateral, low-density GGO, bronchial dilatation, distortion in the lingula, and diffuse subpleural curvilinear opacities. (**C**) A follow-up CT scan after a further 9 months, demonstrates the complete resolution of GGO and minimal residual, subpleural curvilinear opacities.

**Table 1 jcm-11-03577-t001:** Clinical and laboratory characteristics of the patients (baseline and time-course).

	Patient 1	Patient 2
Age	60	66
Smoker	Yes	Yes
BMI (Body mass index)	23.8	26.3
Major comorbidities		
Arterial hypertension	Yes	Yes
Dyslipidemia	Yes	Yes
Other	No	No
PaO_2_/FiO_2_ (mmHg) T = 0	54.8	61.7
PaO_2_/FiO_2_ (mmHg) T = 24 h	62.4	72.9
PaO_2_/FiO_2_ (mmHg) T = 48 h	106	98
PaO_2_/FiO_2_ (mmHg) T = 96 h	136	152
Lung compliance (mL/cmH_2_O) T = 0	22	35
Lung compliance (mL/cmH_2_O) T = 24 h	28	41
C-reactive protein (mg/L) T = 0	283	241
C-reactive protein (mg/L) T = 24	191	232
C-reactive protein (mg/L) T = 48	87	138
C-reactive protein (mg/L) T = 96	23	37
Time to extubation (days)	13	11
Time to weaning from ECMO (Extracorporeal membrane oxygenation) (days)	6	6
Time to discharge (days)	41	38

## Data Availability

Raw data are available upon request to the corresponding author.

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
