# Peer review of "Severe COVID-19 ARDS Treated by Bronchoalveolar Lavage with Diluted Exogenous Pulmonary Surfactant as Salvage Therapy: In Pursuit of the Holy Grail?"

_jcm, 2022, doi:10.3390/jcm11133577_

Round 1
Reviewer 1 Report
1. Why authors focused only on the ATII cells? There are other cellular subsets also that are playing an important role during the COVID-19. Please justify and discuss.
2. Curosurf is a natural porcine lung surfactant. What are the possibilities for the use of bacterially produced or other surfactants? Please discuss.
3. Both patients were on steroids (methylprednisolone) since a hospital admission, then how authors make sure that Curosurf therapy was helping or beneficial? Is there any control group of patients? Please provide the details or justify.
4. Authors should include a flow diagram, showing the whole procedure (timeline and doses) of the Curosurf therapy to make it more reader-friendly.
5. Authors should add a separate section as “Limitations of the study”.
Author Response
R: We thank the reviewer for the comments and the editor for the possibility to ameliorate our manuscript
- Why authors focused only on the ATII cells? There are other cellular subsets also that are playing an important role during the COVID-19. Please justify and discuss.
R: In agreement with the reviewer, we justify and discuss that we focused only on the ATII cells in the paper.
ATII cells are unique in their role of synthesizing and assembling all surfactant components. Furthermore, we
add this paragraph: “The cell-to-cell transmission from ATII to ATI pneumocytes, with viral replication, can
rapidly diffuse the damage to the endothelium. The dysregulated inflammatory response causes the cytokine
storm, the hallmark of the most severe SARS-CoV-2-related ARDS, this causes fibrinogen and other plasma
proteins to leak into the alveoli, where they impair the potentiality the surfactant has to adsorb to the surface and
lower surface tension. The migration of fibroblast and inflammatory cells into the lumen at the alveolar level
causes appositional atelectasis and a loss of gas exchange units. The ATII cell loss means a loss of progenitors for
ATI cells, thus also impairing ATI regenerative capacity”.
- Curosurf is a natural porcine lung surfactant. What are the possibilities for the use of bacterially produced or other surfactants? Please discuss.
Furthermore, there are different types of surfactant (e.g. natural bovine and porcine surfactant and synthetic surfactant). A multicenter study showed that early administration of Surfactant-BL (bovine lung extract surfactant) led to reduced mortality in cardiac patients who developed ARDS postoperatively [40-44]. On the contrary, previous randomized multicenter trial failed to demonstrate any improvement in mortality and oxygenation following the bolus administration of exogenous natural porcine surfactant to patients with ALI/ARDS. One trial showed that synthetic surfactant containing rSP-C had no clinical benefit to patients with severe direct lung injury [40-44]. Several meta-analysis showed that all preparations of surfactant similarly failed to reduce mortality. There were insufficient data available for analysis of changes in oxygenation and ventilation characteristics. We decided to use Curosurf, which is a natural porcine surfactant, because it is the only one marketed in Italy. Furthermore, as reported above, we already used Curosulf for another patient, obtaining a good result and therefore we decided to use it in these two COVID-19 patients.
- Both patients were on steroids (methylprednisolone) since a hospital admission, then how authors make sure that Curosurf therapy was helping or beneficial? Is there any control group of patients? Please provide the details or justify. In agreement with reviewer’s comments we underline in the manuscript that both patients were undergoing 80 mg/day methylprednisolone, since hospital admission, that is for over 10 days
before administering the surfactant therapy, whitout any sign of improvement, meanwhile only 24 hours after surfactant treatment, the PaO2/FiO2 ratio increased: “Both patients were invasively ventilated under volume controlled mode during the whole procedure and remained hemodynamically
stable. Both patients were undergoing 80 mg/day methylprednisolone (nonpatented drug, ATC code
H02AB04) treatment, according to a previously published protocol, since hospital admission.
In particular, it is important to note that steroids were given for over 10 days before administering the surfactant therapy, whitout any sign of improvement, meanwhile only 24 hours after surfactant treatment, the PaO2/FiO2 ratio increased [32].
They did not undergo any other antiviral or experimental treatment for
COVID-19 during the whole hospital stay. Both patients were male (60-year-old and 66-year-old,
respectively), active smokers, non-obese (BMI 23.8 and 26.3, respectively), affected by arterial
hypertension and dyslipidemia but no other major comorbidities. There were no other overlapping
clinical conditions but COVID-19-related ARDS that could explain the acute respiratory failure.
We focused on the change in lung compliance, oxygenation by pO2/FiO2 index and radiological
evolution after surfactant therapy (Table 1). 24 hours after surfactant treatment, the PaO2/FiO2 ratio
increased from 54.8 to 62.4 in one case and from 61.7 to 72.9 in the other, despite a volume overload of
approximately 320 ml. 48 hours after treatment, the PaO2/FiO2 ratio increased to 106 and 98,
respectively. Four days after surfactant treatment the PaO2/FiO2 ratio had increased to 136 and 152,
respectively. Lung compliance, measured through an inspiratory pause during invasive mechanical
ventilation and curarization, raised from 22 to 35 cmH2O in one case and from 28 to 41 cmH2O in the
other after 24 hours from surfactant instillation. In both cases, ECMO support was stopped six days
after surfactant instillation”.
- Authors should include a flow diagram, showing the whole procedure (timeline and doses) of the Curosurf therapy to make it more reader-friendly.
R: In agreement with the reviewer’s observation, we add Figure 1 in the manuscript.
- Authors should add a separate section as “Limitations of the study”.
R: In agreement with the reviewer’s observation, we add Figure 1 in the manuscript.
“6.1 Limitations of the study.
There were some limitations to be noted in our study. First, surfactant delivery in our study was performed
in a single-center, following a uniform protocol with meticulous delivery and retention in the lungs. Second, the
major limitation of our study is the small sample size and the absence of controls, which were difficult to
enroll, since the extreme severity of the disease of the two patients that underwent surfactant treatment.
It was unfortunate that there was insufficient power to statistically confirm the therapeutic role of BAL with
diluted surfactant followed by a low-dose bolus in patients with ARDS. Finally, the selection of patients was
influenced by the availability of resources, including Curosurf therapy during the pandemic”.

Reviewer 2 Report
The manuscript “Severe COVID-19 ARDS treated by bronchoalveolar lavage with diluted exogenous pulmonary surfactant as salvage therapy: in pursuit of the holy grail?” from Barbara Ruaro et al. Studies two cases of COVID-19-ARDS patients who have been successfully treated with diluted surfactant by bronchoalveolar lavage followed by a low-dose bolus of surfactant, however, there are several concerns on this manuscript:
1. If possible, the authors should increase more patients for assessing the efficacy of the protocol, such as different age, gender, underlying disease.
2. Given the success cases, could the authors describe when should be the perfect time and to treat COVID-19-ARDS patients with diluted surfactant by bronchoalveolar lavage. Is this diluted exogenous pulmonary surfactant
are good for all COVID-19-ARDS patients? Or need change the dose based on different situation?
3. Could authors comment on the detail mechanism of treatment with diluted exogenous pulmonary surfactant on COVID-19-ARDS patients (such as the function of cells, cytokines, etc )

Author Response
The manuscript “Severe COVID-19 ARDS treated by bronchoalveolar lavage with diluted exogenous pulmonary surfactant as salvage therapy: in pursuit of the holy grail?” from Barbara Ruaro et al. Studies two cases of COVID-19-ARDS patients who have been successfully treated with diluted surfactant by bronchoalveolar lavage followed by a low-dose bolus of surfactant, however, there are several concerns on this manuscript:
R: We thank the reviewer for the comments and the editor for the possibility to ameliorate our manuscript
- If possible, the authors should increase more patients for assessing the efficacy of the protocol, such as different age, gender, underlying disease.
R: As it is not currently possible for us to increase the study population, in agreement with the reviewer’s observation, we add this paragraph with these limitations in the manuscript.
“6.1 Limitations of the study.
There were some limitations to be noted in our study. First, surfactant delivery in our study was performed
in a single-center, following a uniform protocol with meticulous delivery and retention in the lungs. Second, the
major limitation of our study is the small sample size and the absence of controls, which were difficult to
enroll, since the extreme severity of the disease of the two patients that underwent surfactant treatment.
It was unfortunate that there was insufficient power to statistically confirm the therapeutic role of BAL with
diluted surfactant followed by a low-dose bolus in patients with ARDS. Finally, the selection of patients was
influenced by the availability of resources, including Curosurf therapy during the pandemic”.
- Given the success cases, could the authors describe when should be the perfect time and to treat COVID-19-ARDS patients with diluted surfactant by bronchoalveolar lavage. Is this diluted exogenous pulmonary surfactant are good for all COVID-19-ARDS patients? Or need change the dose based on different situation?
R: Our experience regarding two cases with an extreme severe disease. As reported in the manuscript we treated in the past only another patient affected by ARDS and Wegener who benefited by aforementioned surfactant treatment. We decided to used surfactant also in agreement to previous literature data. We has been underlined all these information in the manuscript and we add Figure 1 : “According to previous literature data, we chose a commercial natural surfactant that is routinely used in premature infants (Curosurf, Chiesi farmaceutici SpA, Parma, Italy) [20, 28]. Moreover our decision was supported by the previous positive experience with a young patient affected by ARDS and Wegener who benefited by aforementioned surfactant treatment. Ten vials of Curosurf (each one containing 3 ml of 80 mg/ml surfactant solution, total amount 2400 mg) were diluted with 500 ml Ringer solution (final phospholipid concentration 4.8 mg/ml). The first 250 ml were instilled in the right main bronchus in 5 aliquots of 50 ml during bronchoscopy and suction was applied after administration. This procedure was repeated five times and took about 10 minutes while the patient was supine. The left lung was then treated similarly; 207 ml were recovered from both lungs (41.4% of the administered volume). After the lavage procedure, a further bolus of 600 mg Curosurf was delivered in each main bronchus and no suctioning was performed after this instillation. The total administered dose of surfactant was 40 mg/kg BW (Figure 1)”.
- Could authors comment on the detail mechanism of treatment with diluted exogenous pulmonary surfactant on COVID-19-ARDS patients (such as the function of cells, cytokines, etc )
In agreement with reviewer’s comments we added this paragraph in the manuscript” The cell-to-cell transmission from ATII to ATI pneumocytes, with viral replication, can rapidly diffuse the damage to the endothelium. The dysregulated inflammatory response causes the cytokine storm, the hallmark of the most severe SARS-CoV-2-related ARDS, this causes fibrinogen and other plasma proteins to leak into the alveoli, where they impair the potentiality the surfactant has to adsorb to the surface and lower surface tension. The migration of fibroblast and inflammatory cells into the lumen at the alveolar level causes appositional atelectasis and a loss of gas exchange units. The ATII cell loss means a loss of progenitors for ATI cells, thus also impairing ATI regenerative capacity”.

Reviewer 3 Report
I read with great interest the manuscript by Ruaro et al., who described the effects of therapeutic bronchoalveolar lavage with diluted surfactant replacement in two patients affected by severe COVID-19 pneumonia.
Major comments:
First of all, the authors present a case report about only two patients with only limited information about their medical history. Moreover, the potential benefit of surfactant administration in COVID-19 patients has already been published several times before. Since then, several groups have initiated trials to investigate the potential benefit of exogenous surfactant administration in patients with COVID-19. The authors give sufficient information about timing, surfactant preparation, dose and delivery technique. However, additional information regarding labaratory parameters, ventilator settings and other rescue maneuvers (i.e. pronation or iNO) before, during or after surfactant treatment would be desirable.
Minor comments:
I would recommend an additional table about basal characteristics and an addtional figure about the main outcome parameters
Please check the spelling and grammar throughout the manuscript:
Page 1 affilliation: giving the institutional email instead private one of the corresponding author is recommended
Page 3 spelling: “demonstrated the lack of” instead of “demonstreted the ipoproduction“
Page 4 wording: the phrase „the last guidelines“ should be replaced by „the latest guidelines“
Page 4: Please specify the paragraph about the noninvasive respiratory support (NIRS). The current wording suggests a recommendation for non-invasive ventilation „instead“ of an invasive one. In fact, however, a necessary intubation should not be delayed. For adults with COVID-19 and acute hypoxemic respiratory failure despite conventional oxygen therapy, starting a high-flow nasal cannula (HFNC) oxygen therapy is recommended; if patients fail to respond, intubation and mechanical ventilation should be initiated.
Page 4: please reformulate „the nasty continuation“
Page 5: reformulate the ending oft he first paragraph , e.g.: „no evidence-based proof for surfactant administration“
Page 5 spelling: „COVID-19-related ARDS“ instead of „COVI-19-realted ARDS“
Page 5: lung compliance is measured in ml/cmH2O
Page 6: please define the abbreviation “GGO” at the first mention in the text
Page 7: How can a “reduction of ventilation duration” be observed from a case report of only two patients? What is the reference for this?
Page 7: How can the conclusion of an “individualized ventilatory approach” can be drawn from an drug intervention? The ventilatory approach was not discussed before.
Author Response
I read with great interest the manuscript by Ruaro et al., who described the effects of therapeutic bronchoalveolar lavage with diluted surfactant replacement in two patients affected by severe COVID-19 pneumonia.
R: We thank the reviewer for the comments and the editor for the possibility to ameliorate our manuscript
Major comments:
First of all, the authors present a case report about only two patients with only limited information about their medical history. Moreover, the potential benefit of surfactant administration in COVID-19 patients has already been published several times before. Since then, several groups have initiated trials to investigate the potential benefit of exogenous surfactant administration in patients with COVID-19. The authors give sufficient information about timing, surfactant preparation, dose and delivery technique. However, additional information regarding labaratory parameters, ventilator settings and other rescue maneuvers (i.e. pronation or iNO) before, during or after surfactant treatment would be desirable.
R: We thank the reviewer for the comments. In agreement with the reviewer, we add Figure 1, Table 1 and this paragraph have been added in the manuscript: “
Table 1. Clinical and laboratory characteristics of the patients (baseline and time-course) |
|||||
Patient 1 |
Patient 2 |
||||
Age |
60 |
66 |
|||
Smoker |
Yes |
Yes |
|||
BMI |
23.8 |
26.3 |
|||
Major comorbidities |
|||||
Arterial hypertension |
Yes |
Yes |
|||
Dyslipidemia |
Yes |
Yes |
|||
Other |
No |
No |
|||
PaO2/FiO2 (mmHg) T=0 |
54.8 |
61.7 |
|||
PaO2/FiO2 (mmHg) T=24h |
62.4 |
72.9 |
|
||
PaO2/FiO2 (mmHg) T=48h |
106 |
98 |
|
||
PaO2/FiO2 (mmHg) T=96h |
136 |
152 |
|
||
Lung compliance (mL/cmH2O) T=0 |
22 |
35 |
|
||
Lung compliance (mL/cmH2O) T=24h |
28 |
41 |
|
||
C-reactive protein (mg/L) T=0 |
283 |
241 |
|
||
C-reactive protein (mg/L) T=24 |
191 |
232 |
|
||
C-reactive protein (mg/L) T=48 |
87 |
138 |
|
||
C-reactive protein (mg/L) T=96 |
23 |
37 |
|
||
Time to extubation (days) |
13 |
11 |
|
||
Time to weaning from ECMO (days) |
6 |
6 |
|
||
Time to discharge (days) |
41 |
38 |
|
Both patients were invasively ventilated under volume controlled mode, with a tidal volume of 6 ml/kg, according to the current standard for protective ventilation in ARDS, during the whole procedure and remained hemodynamically stable. Both patients were undergoing 80 mg/day methylprednisolone (nonpatented drug, ATC code H02AB04) treatment, according to a previously published protocol, since hospital admission [32]. They did not undergo any other antiviral or experimental treatment for COVID-19 during the whole hospital stay. The two patients underwent cycles of prone position until they were wears from invasive mechanical ventilation.
Both patients were male (60-year-old and 66-year-old, respectively), active smokers, non-obese (BMI 23.8 and 26.3, respectively), affected by arterial hypertension and dyslipidemia but no other major comorbidities. There were no other overlapping clinical conditions but COVID-19-related ARDS that could explain the acute respiratory failure. We focused on the change in lung compliance, oxygenation by pO2/FiO2 index and radiological evolution after surfactant therapy (Table 1). 24 hours after surfactant treatment, the PaO2/FiO2 ratio increased from 54.8 to 62.4 in one case and from 61.7 to 72.9 in the other, despite a volume overload of approximately 320 ml. 48 hours after treatment, the PaO2/FiO2 ratio increased to 106 and 98, respectively. Four days after surfactant treatment the PaO2/FiO2 ratio had increased to 136 and 152, respectively. Lung compliance, measured through an inspiratory pause during invasive mechanical ventilation and curarization, raised from 22 to 35 cmH2O in one case and from 28 to 41 cmH2O in the other after 24 hours from surfactant instillation. In both cases, ECMO support was stopped six days after surfactant instillation.
Minor comments:
I would recommend an additional table about basal characteristics and an addtional figure about the main outcome parameters
R: In agreement with the reviewer’s observations, we add Table 1 and Figure 1.
Please check the spelling and grammar throughout the manuscript:
R: In agreement with the reviewer’s observations, we check the spelling and grammar throughout the manuscript
Page 1 affilliation: giving the institutional email instead private one of the corresponding author is recommended
R: In agreement with the reviewer’s the institutional email of corresponding author has been added
Page 3 spelling: “demonstrated the lack of” instead of “demonstreted the ipoproduction“
R: In agreement with the reviewer’s observations, we changed the spelling
Page 4 wording: the phrase „the last guidelines“ should be replaced by „the latest guidelines“
R: In agreement with the reviewer’s observations, we changed the spelling
Page 4: Please specify the paragraph about the noninvasive respiratory support (NIRS). The current wording suggests a recommendation for non-invasive ventilation „instead“ of an invasive one. In fact, however, a necessary intubation should not be delayed. For adults with COVID-19 and acute hypoxemic respiratory failure despite conventional oxygen therapy, starting a high-flow nasal cannula (HFNC) oxygen therapy is recommended; if patients fail to respond, intubation and mechanical ventilation should be initiated.
R: In agreement with the reviewer’s observations, we changed the paragraph: “The latest guidelines for the hospital care of patients affected by coronavirus disease-2019 (COVID-19)-related acute respiratory failure have moved towards a widely accepted use of non-invasive respiratory support (NIRS) in the initial stages of disease. The establishment of severe COVID-19 pneumonia goes through different pathophysiological phases that partially resemble typical ARDS and have been categorized into different clinical-radiological phenotypes. These can variably benefit on the application of external positive end-expiratory pressure (PEEP) during noninvasive mechanical ventilation, mainly due to variable levels of lung recruit ability and lung compliance during different phases of the disease. A growing body of evidence suggests that intense respiratory effort producing excessive negative pleural pressure swings plays a critical role in the onset and progression of lung and diaphragm damage in patients treated with noninvasive respiratory support. Routine respiratory monitoring is mandatory to avoid the continuation of NIRS in patients who are at higher risk for respiratory deterioration and could benefit from non invasive mechanical ventilation and/or early initiation of invasive mechanical ventilation”.
Page 4: please reformulate „the nasty continuation“
R: In agreement with the reviewer’s observations, we reformulate all the paragraph “The latest guidelines for the hospital care of patients affected by coronavirus disease-2019 (COVID-19)-related acute respiratory failure have moved towards a widely accepted use of non-invasive respiratory support (NIRS) in the initial stages of disease. The establishment of severe COVID-19 pneumonia goes through different pathophysiological phases that partially resemble typical ARDS and have been categorized into different clinical-radiological phenotypes. These can variably benefit on the application of external positive end-expiratory pressure (PEEP) during noninvasive mechanical ventilation, mainly due to variable levels of lung recruit ability and lung compliance during different phases of the disease. A growing body of evidence suggests that intense respiratory effort producing excessive negative pleural pressure swings plays a critical role in the onset and progression of lung and diaphragm damage in patients treated with noninvasive respiratory support. Routine respiratory monitoring is mandatory to avoid the continuation of NIRS in patients who are at higher risk for respiratory deterioration and could benefit from non invasive mechanical ventilation and/or early initiation of invasive mechanical ventilation”.
Page 5: reformulate the ending oft he first paragraph , e.g.: „no evidence-based proof for surfactant administration“
R: In agreement with the reviewer’s observations, we reformulate the paragraph” These studies included the administration of the exogenous pulmonary surfactant in adults with acute lung injury and ARDS. These studies have demonstrated beneficial effects in adults affected by ARDS by exogenous surfactant on gas exchange and lung mechanics, but there is no evidence-based proof for surfactant administration [10,16-19]”.
Page 5 spelling: „COVID-19-related ARDS“ instead of „COVI-19-realted ARDS“
R: In agreement with the reviewer ’s observations, we checked the spelling
Page 5: lung compliance is measured in ml/cmH2O
R: In agreement with the reviewer’s observations, we checked the spelling
Page 6: please define the abbreviation “GGO” at the first mention in the text
R: In agreement with the reviewer observations, we defined the abbreviation “GGO”
Page 7: How can a “reduction of ventilation duration” be observed from a case report of only two patients? What is the reference for this?
R: In agreement with the reviewer observations, we clarified in the manuscript that both patients
were undergoing 80 mg/day methylprednisolone, since hospital admission, that is for over 10 days
before administering the surfactant therapy, whitout any sign of improvement, meanwhile only 24
hours after surfactant treatment, the PaO2/FiO2 ratio increased: “Both patients were invasively
ventilated under volume controlled mode during the whole procedure and remained hemodynamically
stable. Both patients were undergoing 80 mg/day methylprednisolone (nonpatented drug, ATC code
H02AB04) treatment, according to a previously published protocol, since hospital admission.
In particular, it is important to note that steroids were given for over 10 days before administering the surfactant therapy, whitout any sign of improvement, meanwhile only 24 hours after surfactant treatment, the PaO2/FiO2 ratio increased [32].
They did not undergo any other antiviral or experimental treatment for
COVID-19 during the whole hospital stay. Both patients were male (60-year-old and 66-year-old,
respectively), active smokers, non-obese (BMI 23.8 and 26.3, respectively), affected by arterial
hypertension and dyslipidemia but no other major comorbidities. There were no other overlapping
clinical conditions but COVID-19-related ARDS that could explain the acute respiratory failure.
We focused on the change in lung compliance, oxygenation by PaO2/FiO2 index and radiological
evolution after surfactant therapy (Table 1). 24 hours after surfactant treatment, the PaO2/FiO2 ratio
increased from 54.8 to 62.4 in one case and from 61.7 to 72.9 in the other, despite a volume overload of
approximately 320 ml. 48 hours after treatment, the PaO2/FiO2 ratio increased to 106 and 98,
respectively. Four days after surfactant treatment the PaO2/FiO2 ratio had increased to 136 and 152,
respectively. Lung compliance, measured through an inspiratory pause during invasive mechanical
ventilation and curarization, raised from 22 to 35 cmH2O in one case and from 28 to 41 cmH2O in the
other after 24 hours from surfactant instillation. In both cases, ECMO support was stopped six days
after surfactant instillation”.
Page 7: How can the conclusion of an “individualized ventilatory approach” can be drawn from an drug intervention? The ventilatory approach was not discussed before.
R: In agreement with the reviewer’s observations, we reformulate the paragraph” 7. Conclusions
In conclusion, therapeutic BAL with diluted surfactant followed by a low-dose bolus seems to be an effective, feasible and safe method of treating patients with ARDS from different causes. It should be emphasized that the lavage technique with diluted surfactant provides a method of selective and direct drug administration together with substantial removal of airway and alveolar debris. Furthermore, the method appears to be very cost-effective. We think that it is the removal of foreign protein, cellular breakdown products, blood, bacteria and mucus that enhances the therapeutic efficacy of the surfactant and thereby enhances ventilation and gas exchange. We therefore suggest that our case reports could be used to generate hypotheses for future prospective controlled studies, which may prove that diluted surfactant lavage is effective and improves ARDS outcome. For this reason, a therapeutic strategy in patients with SARS-CoV-2 infection is crucial. An individualized approach based on lung physiology, morphology, imaging, and identification of biological phenotypes may improve COVID-19 outcome. Indeed, the novelty of our study is that we add surfactant therapy in patients with critical COVID-19 related ARDS, opting for a higher dosage than the one reported in the trials, and using the bronchoscope to deposit the surfactant as far as possible in order to get it to the areas where the damage is most relevant and where there is evidence of alveolar collapse. Further research is needed to further analyze the use of the bronchoscope to deposit the surfactant, with the purpose of optimizing the management of patients with ARDS due to COVID-19”.

Round 2
Reviewer 1 Report
Authors successfully responded to the reviewer's comments/suggestions.